# Targeting COVID-19 Vaccine Hesitancy in Minority Populations in the US: Implications for Herd Immunity

**DOI:** 10.3390/vaccines9050489

**Published:** 2021-05-11

**Authors:** James E. K. Hildreth, Donald J. Alcendor

**Affiliations:** 1Center for AIDS Health Disparities Research, Department of Microbiology and Immunology, School of Medicine, Meharry Medical College, 1005 Dr. D.B. Todd Jr. Blvd., Nashville, TN 37208-3599, USA; jhildreth@mmc.edu; 2Department of Internal Medicine, School of Medicine, Meharry Medical College, 1005 Dr. D.B. Todd Jr. Blvd., Nashville, TN 37208-3599, USA; 3Center for AIDS Health Disparities Research, Department of Microbiology, Immunology, and Physiology, School of Medicine, Meharry Medical College, 1005 Dr. D.B. Todd Jr. Blvd., Nashville, TN 37208, USA

**Keywords:** health disparities, coronavirus, vaccine hesitancy, SARS-CoV-2, COVID-19, minorities populations, health inequities

## Abstract

There has been a continuous underrepresentation of minorities in healthcare research and vaccine trials, along with long-standing systemic racism and discrimination that have been fueling the distrust of the healthcare system among these communities for decades. The history and legacy of racial injustices and negative experiences within a culturally insensitive healthcare system have greatly contributed to vaccine hesitancy among ethnic minorities. COVID-19 vaccine hesitancy will impact vaccine uptake in the US, subsequently hindering the establishment of herd immunity (75–85% of the population vaccinated) to mitigate SARS-CoV-2 infection and transmission. Information targeting underserved racial/ethnic minorities in the US in a culturally competent manner has been lacking. This information is crucial for educating these communities about COVID-19 vaccines and their distribution as well as dispelling misinformation regarding vaccine trials, safety, and efficacy. This lack of education has greatly contributed to COVID-19 vaccine hesitancy and will increase disparities in vaccine uptake. Moreover, timely vaccinations are also essential to curtailing virus transmission and the emergence of SARS-CoV-2 variants that may evade the immune response produced by the three existing COVID-19 vaccines.

## 1. Introduction

SARS-CoV-2 that causes COVID-19 disproportionately affects racial and ethnic minorities compared to non-Hispanic whites (NHW) [1,2,3,4,5,6,7]. Underserved populations who live in poverty with limited access to social services across the US are more likely to have underlying medical conditions [8,9]. Minorities are also among the most vulnerable to the more severe complications of SARS COV-2. COVID-19 vaccine hesitancy is one of the greatest challenges that we must overcome in the US [10]. Vaccine hesitancy is among the top ten global health threats identified by the World Health Organization in 2019 [11]. This high level of hesitancy is observed in minority communities in light of FDA emergency use authorization of three different vaccines from Pfizer/BioNtech, Moderna, and Johnson & Johnson-Janssen-Pharmaceuticals (J&J) that have been shown in Phase III clinical trials to be immunogenic, safe, and efficacious [12,13,14]. In a study by Coustasse et al., only 25% of AA and 37% of H/L would get a COVID-19 vaccine, compared to 56% of NHW [15]. Experience with racial injustice within the medical establishment has contributed greatly to widespread COVID-19 vaccine hesitancy among AA and H/L [16]. Societal and cultural barriers that prevent ethnic minorities from achieving health equity, including vaccine access, are systemic issues that may be addressed only through shifts in governmental policies that can effect substantial changes to combat healthcare inequities. The success of COVID-19 vaccines hinges on high vaccine acceptance and strategies to combat misinformation promoted by the antivax movement [17,18]. Vaccine hesitancy, misinformation campaign by antivaxxers, and general distrust of the medical establishment could reduce vaccine uptake to a level below that required for establishing herd immunity (Figure 1). This review explores various issues and consequences associated with vaccine hesitancy among minority populations in the US, as well as strategies for combating vaccine hesitancy and promoting equity in COVID-19 vaccination.

## 2. The Safety and Efficacy of COVID-19 Vaccines

Operation Warp Speed (OWS) was a partnership initiated by the US government with the private sector to facilitate and accelerate the development, manufacturing, and distribution of COVID-19 vaccines, therapeutics, and diagnostics [19,20]. Entities that received direct government funding include J&J, AstraZeneca-University of Oxford, Moderna, Novavax, Merck-IAVI, Sanofi, and GlaxoSmithKline. While Pfizer/BioNtech did not receive government funding, it participated in OWS as a COVID-19 vaccine supplier. At the time of the writing of this review, COVID-19 vaccines from Moderna, Pfizer/BioNtech, and J&J have received emergency use authorization (EUA) in the US from the FDA [21] (Table 1).

### 2.1. The Pfizer/BioNtech COVID-19 Vaccine (BNT162b2)

The Pfizer vaccine is a lipid nanoparticle-formulated, nucleoside-modified mRNA vaccine that encodes the prefusion spike glycoprotein of SARS-CoV-2 (Wuhan 2019; whole genome sequence, NC_045512) [22]. In a Phase 3 clinical trial, 43,548 participants aged 16 years and older were randomized to receive two 30-µg doses of the vaccine or a saline placebo 21 days apart [22] (Table 1). The vaccine showed an efficacy of 95% against symptomatic COVID-19, with eight cases reported in the vaccine group and 162 cases in the placebo group [22] (Table 1). The most common adverse events were mild-to-moderate pain at the injection site, fatigue, and headache. These adverse events were more common in younger participants and occurred more frequently after the second dose [22,23].

### 2.2. The Moderna COVID-19 Vaccine (mRNA-1273)

The Moderna vaccine is also a lipid nanoparticle-formulated, nucleoside-modified mRNA vaccine that encodes the prefusion spike glycoprotein of SARS-CoV-2 (Wuhan 2019; whole genome sequence, NC_045512) [24,25]. In a Phase 3 clinical trial, 30,420 participants aged 18 years and older were randomized to receive two 100-µg doses of the vaccine or a saline placebo 28 days apart [24] (Table 1). The vaccine showed an efficacy of 94.1% against symptomatic COVID-19, with 11 cases reported in the vaccine group and 185 cases in the placebo group. Adverse event occurrence in the vaccine and placebo groups were 84.2% and 19.8%, respectively, for the first dose and 88.6% and 18.8%, respectively, for the second dose [25] (Table 1). Adverse events were more common in the younger participants (aged 18–65 years) than in the older participants (aged 65 years and above).

### 2.3. The J&J COVID-19 Vaccine (JNJ-78436735/Ad26.COV2.S)

The single-dose Johnson & Johnson’s Janssen (J&J/Janssen) COVID-19 vaccine consists of a recombinant, replication-incompetent human adenovirus serotype 26 (Ad26) vector that encodes a full-length and stabilized SARS-CoV-2 spike protein [26] (Table 1). The spike glycoprotein sequence used is from the first clinical isolate of the Wuhan strain (Wuhan 2019; whole genome sequence, NC_045512) [27]. In a Phase 3 clinical trial in the US, 43,783 participants aged 16 years and older were randomized to receive a 0.5-mL dose containing 5 × 10^10^ virus particles or a placebo. The vaccine showed an efficacy of 72% against symptomatic COVID-19 with 85% efficacy overall against severe disease. Common adverse events included injection site pain, fever, fatigue, headache, and myalgia [28].

### 2.4. Vaccine Safety

FDA EUA guidance for both the Pfizer and Moderna vaccines is to not administer the vaccines to individuals with a known history of a severe allergic reaction (e.g., anaphylaxis) to any component of the COVID-19 vaccines. In addition, persons with a history of anaphylaxis should be observed for 30 min rather than 15 min after COVID-19 vaccinations for both Pfizer and Moderna vaccines [29]. Even more, leading women’s health care organizations and government health authorities have recommended that Pfizer and Moderna COVID-19 vaccines be offered to those women who are pregnant and or breastfeeding because the potential benefits of maternal vaccination during pregnancy and or lactation outweigh the risks [30]. In a study performed by Shimabukuro et al., who examined 35,691 women v-safe participants 16 to 54 years of age that identified as pregnant, injection site pain was the most frequently reported by pregnant women compared to non-pregnant women. Among 3958 pregnant women enrolled in the v-safe pregnancy registry, 827 had a completed pregnancy, of which 115 (13.9%) experienced pregnancy loss and 712 (86.1%) resulted in a live birth (mostly among participants with vaccination in the third trimester) [31]. Although no neonatal deaths were reported, adverse neonatal outcomes included preterm birth (in 9.4%) and small size for gestational age (in 3.2%) [31]. Even more, although not directly comparable, there appeared no significant difference in adverse events and neonatal outcomes compared to other studies with similar cohorts before the COVID-19 pandemic. Anaphylaxis at a rate of 11.1 per million doses has been reported for the Pfizer vaccine [32]. Anaphylaxis has also been reported for the Moderna vaccine; however, the specific mechanism supporting the allergy and the specific antigen has not been identified. Hypersensitivity-related adverse events were observed in 0.63% of Pfizer and 1.5% of Moderna vaccine trial participants when compared to 0.51% and 1.1%, respectively, in the placebo groups [29]. Anaphylaxis after COVID-19 vaccination is rare. Administration of the single-dose Johnson & Johnson’s Janssen (J&J/Janssen) has been associated with adverse events that include pain, redness, and swelling at the site of injection along with fatigue, headaches, muscle pain, chills, fever, and nausea. These common side effects were found to be transient and were gone after a few days. However, a pause was recommended by the FDA and CDC after reports of six cases of a rare and severe type of blood clot in individuals following administration of the Janssen COVID-19 Vaccine was observed. The FDA and CDC examined available data to assess the risk of thrombosis involving the cerebral venous sinuses, or CVST, and other sites in the body that included large blood vessels of the abdomen and the veins of the legs along with thrombocytopenia. Following the review, the FDA and CDC recommended that use of Johnson & Johnson’s Janssen (J&J/Janssen) COVID-19 Vaccine resume in the United States, effective 23 April 2021. However, women younger than 50 years old especially should be aware of the rare risk of blood clots with low platelets after vaccination, and that other COVID-19 vaccines are available where this risk has not been observed.

## 3. Vaccine Hesitancy

Vaccine hesitancy refers to a delay in uptake or refusal of a vaccine even when the vaccine is proven safe and widely accessible [33,34]. Vaccine hesitancy is often complex and can be influenced by various factors pertaining to race, ethnicity, age, gender, income levels, health insurance access, religious and moral convictions, political affiliation, level of education, historical mistrust in the biomedical and healthcare establishment, and overall mistrust of the scientific enterprise of medicine and public health, to name a few [35,36,37,38,39].

## 4. COVID-19 Vaccine Hesitancy among Major Ethnic Groups in the US

### 4.1. Non-Hispanic-Whites

National surveys have shown NHW to be more likely to agree to COVID-19 vaccination than H/L and AA [40]. However, subpopulations of NHW living in rural communities that tend to be less educated, have lower incomes, and identify as Republicans, are generally more reluctant to get vaccinated against COVID-19 [41]. In a study by Doherty et al. that examined racial minorities and marginalized populations across nine counties in North Carolina, the authors found an overall prevalence of vaccine hesitancy of 68.9%, which included 62.7% for NHW, 74% for AA, and 59.5% for H/L [42]. In this region, vaccine hesitancy among NHW is second only to that among AA and significantly higher than that among NHW who also identify as Republicans but have more education and higher incomes. Residents of rural communities, Republicans, and AA are the most hesitant with regards to COVID-19 vaccination [42].

In a survey conducted by the Kaiser Family Foundation COVID-19 Vaccine Monitor project, approximately three in 10 rural residents would accept the vaccine compared to four in 10 in both urban and suburban areas [43]. The same survey also showed that 42% of Republicans and 12% of Democrats as well as 26% of NHW and 35% of AA had indicated that they would probably not or definitely not get the COVID-19 vaccine [43]. Approximately 25% of both urban and suburban residents were hesitant about getting a COVID-19 vaccine [43]. The survey was conducted from 30 November to 8 December 2020 among 1700 adults aged 18 years and older [43].

In a recent poll by the Associated Press-NORC Center for Public Affairs Research, 67% of Americans indicated that they either plan to get vaccinated or have already been vaccinated. However, a significant number of individuals who identify as Republicans and individuals without college degrees are still hesitant about getting the COVID-19 vaccine [44]. This level of hesitancy could greatly impact our efforts to achieve herd immunity (Figure 1).

### 4.2. African American

AA are the most heavily impacted by COVID-19 with the highest percentage of COVID-19 cases, the most severe clinical presentation, and the highest rates of hospitalization and COVID19 morbidity and mortality in both adults and children [45,46,47]. Access to COVID-19 testing, mitigation equipment, and vaccines is also lower among AA [48]. Vaccine administrators often collect limited racial data among vaccine recipients, likely resulting in a skewed observation of the vaccinated population. These are some of the factors that lead to reluctance among many AA to get vaccinated and the exacerbation of existing disparities in COVID-19 outcomes. The erroneous belief among AA that they are not susceptible to SARS-CoV-2 infection because of the melanin in their skin is among the claims that likely cost many lives at the height of the pandemic [49]. Many AA also do not trust the COVID-19 vaccines developed during OWS because they believe that shortcuts were taken during vaccine development and manufacturing. The social, political, and economic injustices that AA have endured for decades reinforce their distrust of the healthcare system. They also constitute a significant barrier against cultivating trust and confidence among AA towards COVID-19 vaccines. Therefore, there is an urgent need for health equity initiatives that can lead to policies that address these problems.

### 4.3. Hispanic/Latinx

H/L in the US have been disproportionately impacted by COVID-19 because they are more likely to be hospitalized and die from COVID-19 than NHW [50]. Historical injustices and misinformation have rendered H/L skeptical of the safety and efficacy of COVID-19 vaccines. Existing disparities in vaccine access and distribution may also contribute to vaccine hesitancy among H/L. The US Centers for Disease Control and Prevention reported that during the first month when vaccines became available, H/L and AA received 11.5% and 5.4% of the available vaccines, respectively, while NHW received over 60% [51]. This jarring racial gap in COVID-19 vaccination continues throughout the US and warrants urgent attention. H/L favor a tight-knit community culture in which they rely on trusted peers and messengers for information to guide their decisions about COVID-19 vaccines. Immigration status also impacts vaccine acceptance among H/L. Undocumented immigrants are reluctant to both test for COVID-19 and receive a COVID-19 vaccine because they want to avoid encounters with immigration authorities. These individuals live in fear for themselves and their families; hence, they will likely remain unvaccinated and transmit the virus to others if they get infected. There is an urgent need for public figures and role models in communities of H/L to help dispel misinformation, misperceptions, conspiracy theories, and myths about COVID-19 and COVID-19 vaccines.

There is also vaccine hesitancy in underserved populations in the developing world that could impact herd immunity on a global scale. A recent survey conducted by CompariSure reported that 52% of South Africans will not take the COVID-19 vaccines with religion, fear of needles, and unconsented government tracking being reported as the main reasons for not accepting the vaccine [52]. A Zimbabwean COVID-19 vaccine hesitancy survey also found that 50% of Zimbabweans would not accept the COVID-19 vaccine [53]. South African have historically been non-accepting of vaccinations in general. Prominent theologians have warned against COVID-19 vaccine acceptance, which will lead to increased infection rates, hospitalizations, and mortality among African nations. Beliefs held by many religious leaders in South Africans have had a negative impact on mass vaccination programs and have greatly reduced public trust in COVID-19 vaccinations [54]. The similarities in COVID-19 vaccine hesitancy are not unlike what is observed in the US. Hesitancy associated with long-standing fear, misinformation, antivaxxer propaganda, religious and political affiliations, and mistrust is share among global communities.

## 5. Hesitancy Associated with Vaccine Supply and Global Inequities

Approximately 330 million doses of the J&J vaccine or 660 million doses of either the Pfizer/BioNTech or Moderna vaccine are needed to vaccinate the entire US population. At the time of writing this review, vaccine supply has fallen short of demand. Vaccine manufacturers have pledged to increase their production capacity or have proposed manufacturing partnerships to increase vaccine supply.

Some forms of vaccine hesitancy are directly linked to vaccine supply shortage. For some in the US who are contemplating COVID-19 vaccination, supply shortage may prompt them to delay or avoid vaccination altogether. For those anxiously awaiting vaccination when vaccine supply is scarce, their frustration with the wait could cause them to give up vaccination. It is unclear how low-income countries around the world will get equitable access to COVID-19 vaccines. Strategies to combat vaccine hesitancy on a global scale in a culturally appropriate manner are essential. The global population is connected through international travel and trade. Therefore, to curb COVID-19 spread, we must vaccinate the global community, including individuals from low-income countries and marginalized populations. To achieve this, standards of practice for fair vaccine accessibility and affordability are essential.

In the US, vaccine supply shortage is especially apparent at vaccination sites that serve minority populations. Additionally, there are fewer vaccination sites and staff members serving minority communities. These factors contribute to inequities in vaccination among underserved communities most heavily impacted by COVID-19. Consequently, individuals in these communities are less equipped to navigate the complexities of vaccine access and more likely hesitant to get vaccinated.

Globally, high-income countries possess 60% of the world’s COVID-19 vaccine supply even though they make up only 17% of the global population. For example, Canada has acquired 300 million doses for its 100 million citizens. Conversely, Nigeria currently does not have access to the vaccines for its 220 million citizens. This inequity in vaccine distribution will perpetuate the pandemic, allow viral variants to emerge, and continue to cripple the global economy. A virtual world summit that meets regularly may be a viable step to identify barriers against global vaccine equity as well as to ensure that low-income countries and marginalized populations have access to COVID-19 vaccines. Raging SARS-CoV-2 infections in developing countries and continual outbreaks in developed countries indicate that COVID-19 is endemic on a global scale. Strategies to facilitate continual vaccinations and vaccine development as well as to curtail the evolution of viral variants are critical.

## 6. Vaccine Hesitancy among Healthcare Providers

Healthcare providers are among the most trusted advisors and influencers of vaccination decisions. Recommendations from physicians correlate strongly with vaccine acceptance among their patients [55]. Unfortunately, many healthcare providers are hesitant to accept COVID-19 vaccines. A study to evaluate COVID-19 vaccine hesitancy among medical students at one medical school in the US showed that while many expressed positive attitudes towards COVID-19 vaccines [56], only 53% indicated that they would participate in COVID-19 vaccine trials. Moreover, 23% indicated that they would not accept a COVID-19 vaccine approved by the FDA [56]. Clinical students were more likely to participate in a vaccine trial (62 vs. 44%, P = 0.02), while students who expressed trust in health experts were more likely to agree to take the vaccine and have fewer concerns about side effects [56]. Limitations of this study include a low response rate (37%) and a small sample size of students from a single medical school [56]. Nonetheless, this study demonstrates the need to incorporate a curriculum designed to enhance the knowledge of medical students about COVID-19 vaccines and to teach them vaccine counseling skills.

Shekar et al. conducted a cross-sectional study in the US to examine COVID-19 vaccine acceptance among healthcare workers [57]. Among the 3479 healthcare workers surveyed, 36% were willing to take the vaccine, while 56% were unsure or hesitant. Only 8% indicated their refusal to get vaccinated [57]. Vaccine acceptance increased with age, education level, and income level. A smaller percentage of AA (19%), Latinx (30%), and rural (26%) healthcare workers were willing to take the vaccine [57]. COVID-19 vaccine safety (69%), effectiveness (69%), and speed of development/approval (74%) were the major concerns of these participants [57].

High levels of vaccine hesitancy have been observed in the southern US. A study by Doherty et al. surveyed multiracial participants from nine counties in North Carolina who represent underserved communities in the state. The authors examined the prevalence of COVID-19 vaccine hesitancy and attitudes regarding vaccinations and assessed their correlation with race [42]. There were 948 participants in this study, including 27.7% Whites, 59.6% Blacks, and 12.7% Latinx. A total of 63% were women. Among these participants, 32% had an income of less than $20,000, 60% owned a computer, and 80% had internet access at home. The overall prevalence of COVID-19 vaccine hesitancy was 68.9%, accounting for 62.7%, 74%, and 59.5% among Whites, Blacks, and Latinx, respectively [42]. Multivariable logistic regression analysis revealed the following factors to be associated with vaccine hesitancy: women (OR = 1.90 95%CI [1.36, 2.64]), Blacks (OR = 1.68 [1.106, 2.45]), calendar month (OR = 0.76 [0.63, 0.92]), safety concerns (OR = 4.28 [3.06, 5.97]), and distrust of the government (OR = 3.57 [2.26, 5.63]) [42].

## 7. The Role of Antivaxxers in Promoting Vaccine Hesitancy

The antivax movement continues to expand globally and is becoming more organized with significant financial support [57,58,59,60]. The movement distorts scientific data and exaggerates documented reports of side effects while minimizing the benefits of vaccinations. Antivaxxers and anti-vaccination activists tend to seek out and believe misinformation regarding vaccine safety while actively trying to reinforce and promote their views about vaccines in irrational ways that can fuel public distrust [61]. Anti-vaccination activists have made many outrageous claims regarding vaccine safety and efficacy. These claims include: (1) vaccines are contaminated with harmful additives; (2) vaccines cause allergies; (3) vaccines can cause autism in children; and (4) vaccines are part of a conspiracy between big pharma and the government to cause harm to the population. Many antivaxxers also view vaccination as an infringement on personal liberty.

By not accepting safe and effective vaccines, antivaxxers can impose undue harm on society [62]. A report by Brennen recommends that governments mandate vaccinations [62]. For instance, the Vaccination Act of 1853 in the UK made vaccination mandatory for all infants within the first three months of life to be vaccinated for smallpox and noncompliant parents liable to fines or imprisonment [63]. Moreover, the Vaccination Act of 1867 required all children under 14 years of age to be vaccinated for smallpox, or their parents would face cumulative penalties [63]. Anti-vaccination activists at the time refused to comply; instead, they claimed that the Act was a violation of their bodies as a form of political tyranny. Mass public demonstrations ensued [64,65]. In 1905, the US Supreme Court ruled that states had the right to install and enforce mandatory vaccination laws [66]. However, compulsory vaccination against COVID-19 seems unlikely [67].

Social media platforms and internet providers can play a role in curbing misinformation regarding COVID-19 vaccine safety and efficacy so that science-based information can reach communities that are vulnerable to misinformation. The benefits of COVID-19 vaccines have been overshadowed by political polarization in unprecedented ways. The antivax movement may lower support for science-based policy interventions and lead to reservoirs of unvaccinated individuals that could continue to transmit viral variants of increased virulence that could become completely resistant to the developing vaccines. Therefore, constructive discussions with antivaxxers are essential to promoting acceptance and uptake of COVID-19 vaccines.

## 8. Strategies Targeting Vaccine Hesitancy to Achieve Equity in COVID-19 Vaccination

The success of COVID-19 vaccines will depend on high vaccine acceptance and strategies that combat the lack of education about vaccines and vaccine misinformation in a culturally competent manner. It is essential to provide minority communities in the US with access to COVID-19 clinical trials and COVID-19 vaccines. Unvaccinated minority communities, including the incarcerated, homeless, and migrant communities, are potential infection reservoirs for COVID-19 that may prolong the pandemic in the US. As a result, SARS-CoV-2 variants that are more virulent or can evade the effects of currently available vaccines may emerge, potentially leading to high rates of mortality and disease with little hope of a return to normalcy. Therefore, it is essential that communities throughout the US have a high rate of vaccine acceptance in order to mitigate COVID-19 transmission in a timely manner.

COVID-19 vaccination among children (up to 17 years of age) in the US is essential to curbing the pandemic. Unfortunately, this strategy is problematic due to vaccine hesitancy among parents. Parents who are hesitant about COVID-19 vaccines are also less likely to allow their children to be vaccinated. According to the 2010 Census, there were 74.2 million people under the age of 18 in the US (24% of the total US population) [68]. This would represent nearly a quarter of the population that would require COVID-19 vaccination. Therefore, vaccination of this portion of the US population is essential to attaining herd immunity (Figure 1). Importantly, children constitute an infection reservoir even though infected children are usually asymptomatic. Comprehensive data on the transmission of viral variants among children are lacking, and new clinical trials that include children are still in their early stages. There is a great need for a national interventional education and awareness campaign that educates parents and children about the importance of getting a COVID-19 vaccine. There is also a great need for more research and larger case-control trials involving children, including those with underlying medical conditions.

Virtual platforms and events that focus on vaccine hesitancy as well as radio commentaries, flyers, and pamphlets in multiple languages, can provide comprehensive outreach to minority communities. Surveys, such as those conducted before and after specific events, and interviews can also help identify barriers against vaccine acceptance. Virtual town hall meetings sponsored by community leaders and local health care providers can engage minority communities in dialogs that address questions these communities have about COVID-19 vaccines.

Minority and marginalized communities in the US, including the medically disabled, homeless, indigenous peoples, and undocumented immigrants, need better access to new and existing COVID-19 preventive infrastructure. We need to deliver the vaccines and other forms of available assistance to individuals who have no access to vaccination sites. Possible strategies to accomplish this include the use of mobile vaccine delivery units and the deployment of vaccine administrators who deliver vaccines to the homeless and other transient and immobile populations.

A trained workforce is essential to timely COVID-19 vaccination in the population. The expansion of vaccination sites to unconventional venues, such as the Department of Motor Vehicles, restaurants, churches, salons, coffee shops, post offices, grocery stores, bus/train stations, theme parks, and shelters, can increase COVID-19 vaccine accessibility. Announcements on social media platforms can help individuals find and make vaccination appointments.

There is also a need for culturally sensitive approaches so that we can identify subpopulations at risk of vaccine hesitancy. Standards of care in underserved minority communities that combat COVID-19 vaccine hesitancy and promote equitable vaccine distribution are warranted. Research in vaccine development and technologies is also crucial to improving COVID-19 vaccine access, particularly in low-income countries with limited infrastructure. For instance, the Pfizer/BioNTech and Moderna vaccines require extreme storage conditions that many low-income countries cannot afford. Therefore, innovations that can improve vaccine accessibility are crucial to ensuring equity in COVID-19 vaccine distribution.

## 9. Conclusions

The level of public trust in vaccines warrants examination in the context of specific populations and their experiences with vaccines. A high level of vaccine acceptance across all populations in the US is essential to mitigating the further transmission of SARS-CoV-2 and its variants as well as to achieving herd immunity (Figure 1). It is also important for the US to lead the way in providing COVID-19 vaccine access to low-income and underserved global communities. The devastations that COVID-19 has inflicted on our communities, economy, and health are a stark reminder that we must always be prepared for future pandemics and that no one is truly isolated in this global community.

This report was not an attempt to quantify COVID-19 vaccine hesitancy in US. This report is an analysis or snapshot of vaccine hesitancy as it exists at the time of this writing among minority communities in the US. We also acknowledge differences among study participants vs. differences in the methods of data collection cited here. Vaccine hesitancy changes with the dynamics of the pandemic, such as access to vaccines, ongoing education and awareness campaigns, and behavioral changes among individuals that are hesitant to accept the vaccine.

## Figures and Tables

**Figure 1 vaccines-09-00489-f001:**
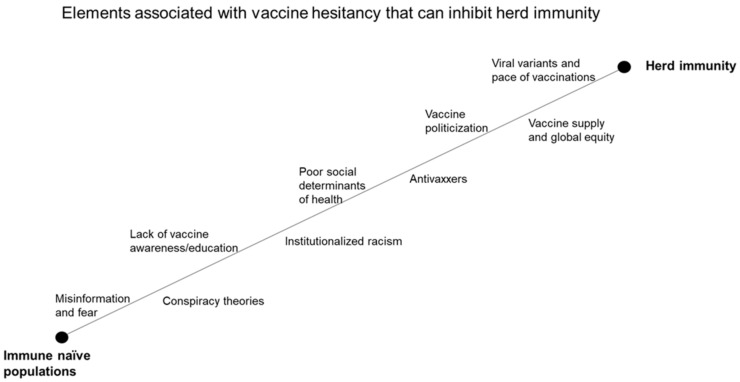
Elements of COVID-19 vaccine hesitancy that can impact herd immunity.

**Table 1 vaccines-09-00489-t001:** SARS-CoV-2 vaccines that received EUA from the FDA.

Developer	Type	Doses	Participants/P3	Study Location	Efficacy/US	EUA Date
Pfizer-BioNTech	mRNA	2	43,548	International	95%	11 December 2020
Moderna-NIAID	mRNA	2	34,420	United States	94%	18 December 2020
Johnson&Johnson-Janssen	Viral vector	1	43,783	International	72%	27 February 2021

EUA (Emergency Use Authorization), NIAID (National Institute of Allergy and Infectious Diseases) P3 (Phase III clinical trials).

## Data Availability

The study did not report any data.

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
