# Peer review of "Targeting COVID-19 Vaccine Hesitancy in Minority Populations in the US: Implications for Herd Immunity"

_vaccines, 2021, doi:10.3390/vaccines9050489_

Round 1

Reviewer 1 Report

My main concerns focus on the structure and on the formal organization of this paper. While sections 1-->5 are clearly consistent with the main title of this paper (i.e. Targeting COVID-19 Vaccine Hesitancy in Minority Populations in the US...", following ones aren't, or at least not totally. I would suggest to either refocus sections 6 to 7 (particularly, section 7) on the minorities or to remove them (in case, it would be preferable to remove Sec. 6 instead of 7). Moreover, please be aware that the text is affected by some redundancies: Authors (see later) discuss the meaning of "Vaccine Hesitancy" in various stages of the paper, repeating its definition. Please report VH definition at its first appearance and stick with it. Moreover, Discussion is somewhat poor in terms of re-working of previous stages, and should be revised accordingly. 

MOREOVER:

  • Table 1 is not self-explanatory. Please include in a note/legend the meaning of the acronyms. Similarly, it is very important to stress that which viral vector is employed by "J&J" vaccine, as similar adenovirus from AstraZeneca and other manufacturers are being implemented. Please also explain how the "efficacy" was calculated. If the efficacy wasn't calculated by Authors, but retrieved from published studies, that should be properly cited
  • I would suggest to amend titles of subsections 4.1 to 4.3 in order to avoid acronyms.
  • The text "At the time of the writing of this review, three COVID-19 vaccines have received EUA from the FDA: one each from Pfizer/BioNTech, Moderna, and J&J. Vaccines from Pfizer/BioNTech and Moderna are two-dose vaccines, while the J&J vaccine is a single- dose vaccine." is again somewhat duplicating previous statements. 
  • When dealing with VH, Authors should state that - as a common and shared framework to characterize and quantify VH doen't exist, the various studies they reported may be difficult to reconciliate, with resulting heterogeneities being difficult to univocally associate with actual differences among study participants vs. differences in the methods to collect their K/A/P. Moreover, K/A/P on COVID-19 and COVID-19 vaccines have exhibited a clear time-dependent trend, and such characteristics should be preventitively discussed.

Author Response

May 4, 2021

Editor and Chief
Journal Vaccines

Manuscript ID: vaccines-1181419

Dear Editor,

My responses to reviewers comments regarding manuscript Manuscript ID: vaccines-1181419 entitled “Targeting COVID-19 Vaccine Hesitancy in Minority Populations in the US: Implications for Herd Immunity” are enclosed.

Thank you for giving me the opportunity to submit this manuscript to your journal for publication.

Kind regards,

Donald J. Alcendor, Ph.D.

Associate Professor

Meharry Medical College

Center for AIDS Health Disparities Research

& Department of Microbiology and Immunology

& Obstetrics and Gynecology

Hubbard Hospital 5th Floor Rm. 5025

1005 Dr. D.B. Todd Jr. Blvd.

Nashville, TN 37208

Phone: 615-327-6449

Fax: 615-327-6929

Email: dalcendor@mmc.edu

Associate Professor Adjunct

Department of Pathology, Microbiology and Immunology

Vanderbilt University Medical Center

Reviewer 1

Comments and Suggestions for Authors

My main concerns focus on the structure and on the formal organization of this paper. While sections 1-->5 are clearly consistent with the main title of this paper (i.e. Targeting COVID-19 Vaccine Hesitancy in Minority Populations in the US...", following ones aren't, or at least not totally. I would suggest to either refocus sections 6 to 7 (particularly, section 7) on the minorities or to remove them (in case, it would be preferable to remove Sec. 6 instead of 7). Moreover, please be aware that the text is affected by some redundancies: Authors (see later) discuss the meaning of "Vaccine Hesitancy" in various stages of the paper, repeating its definition. Please report VH definition at its first appearance and stick with it. Moreover, Discussion is somewhat poor in terms of re-working of previous stages, and should be revised accordingly. 

Authors’ response

We somewhat agree with the reviewer and have removed the last paragraph of Section 6 as it was not related to minority vaccine hesitancy the heath care workplace.  We would prefer to keep the remaining contents of section 6 based on the fact that a highly significant number of essential health care workers are from African American and Hispanic/Latinx communities as shown in the data presented and is highly relevant to the topic of COVID-19 hesitancy in minority communities.  For example, the population of African Americans and Hispanic Latinx in North Carolina is 22.2% and 9.8% respectively. 

There is no Discussion in manuscript only a Conclusion.  We have added language to the Conclusion to address the reviewers’ comments accordingly.

MOREOVER:

  • Table 1 is not self-explanatory. Please include in a note/legend the meaning of the acronyms. Similarly, it is very important to stress that which viral vector is employed by "J&J" vaccine, as similar adenovirus from AstraZeneca and other manufacturers are being implemented. Please also explain how the "efficacy" was calculated. If the efficacy wasn't calculated by Authors, but retrieved from published studies, that should be properly cited.

Authors’ response

We have included meanings of the acronyms in the table (Blue text under the table). We have added information to revised manuscript to address the specifics of the AD vectors employed in the J&J vaccine and is highlighted in blue text in section #3 of the revised manuscript.  The efficacy of the vaccines were provided by published studies and the efficacies are given in the section for the different vaccine for the different Phase 3 clinical trials (Sections 3.1-3.3) of the revised manuscript.

  • I would suggest to amend titles of subsections 4.1 to 4.3 in order to avoid acronyms.

Authors’ response

We agree with the reviewer and have amended the titles to avoid acronyms and these titles appear in blue text in the revised manuscript.

  • The text "At the time of the writing of this review, three COVID-19 vaccines have received EUA from the FDA: one each from Pfizer/BioNTech, Moderna, and J&J. Vaccines from Pfizer/BioNTech and Moderna are two-dose vaccines, while the J&J vaccine is a single- dose vaccine." is again somewhat duplicating previous statements. 

Authors’ response

We agree with the reviewer and have remove this repetitive language from section 5 of the revised manuscript and these revisions appear in blue text.

  • When dealing with VH, Authors should state that - as a common and shared framework to characterize and quantify VH doen't exist, the various studies they reported may be difficult to reconciliate, with resulting heterogeneities being difficult to univocally associate with actual differences among study participants vs. differences in the methods to collect their K/A/P. Moreover, K/A/P on COVID-19 and COVID-19 vaccines have exhibited a clear time-dependent trend, and such characteristics should be preventitively discussed.

We agree with the reviewer and we are aware that it is difficult to quantify vaccine hesitancy. This report was not an attempt to quantify COVID-19 vaccine hesitancy in US.  This report is an analysis or snapshot of vaccine hesitancy as it exists at the time of this writing among minority communities in the US.  Vaccine hesitancy is a moving target that constantly changes with the dynamics of the pandemic such as access to vaccines, ongoing education and awareness, and behavioral changes among individuals that refuse not accept the COVID-19 vaccine.  This pandemic is unprecedented in modern times and the consequences for not reaching herd immunity on a global scale could allow continual transmission of the virus and its variants causing increase mortality and global economic hardships.    We also acknowledge differences among study participants vs. differences in the methods of data collection cited here.  There is language in the Conclusion section of the revised manuscript to address the reviewers’ comments in blue text.

Reviewer 2 Report

This is a very interesting review that explores various issues and consequences associated with vaccine hesitancy among minority populations in the US. Only some minor points should be addressed to improve the manuscript.

- Please add some more data regarding the safety of the vaccines (e.g real world side effects observed from the v-safe application and VAERS, safety in pregnancy and the recent pause of the Janssen vaccine due to VIPIT)

- Some more data from minorities across the world  and the comparison with the US data would be also heplful

-Please remove the parenthesis ‘(Figure 1)’ from the abstract.

Author Response

May 4, 2021

Editor and Chief
Journal Vaccines

Manuscript ID: vaccines-1181419

Dear Editor,

My response to the reviewer’s comments regarding manuscript Manuscript ID: vaccines-1181419 entitled “Targeting COVID-19 Vaccine Hesitancy in Minority Populations in the US: Implications for Herd Immunity” are enclosed. 

Thank you for considering this manuscript submitted to your journal for publication.

Kind regards,

Donald J. Alcendor, Ph.D.

Associate Professor

Meharry Medical College

Center for AIDS Health Disparities Research

& Department of Microbiology and Immunology

& Obstetrics and Gynecology

Hubbard Hospital 5th Floor Rm. 5025

1005 Dr. D.B. Todd Jr. Blvd.

Nashville, TN 37208

Phone: 615-327-6449

Fax: 615-327-6929

Email: dalcendor@mmc.edu

Associate Professor Adjunct

Department of Pathology, Microbiology and Immunology

Vanderbilt University Medical Center

Reviewer 2

Comments and Suggestions for Authors

This is a very interesting review that explores various issues and consequences associated with vaccine hesitancy among minority populations in the US. Only some minor points should be addressed to improve the manuscript.

- Please add some more data regarding the safety of the vaccines (e.g real world side effects observed from the v-safe application and VAERS, safety in pregnancy and the recent pause of the Janssen vaccine due to VIPIT)

Authors’ response

We agree with the reviewer and have added information to the revised manuscript in blue text regarding from the v-safe application and VAERS, safety in pregnancy and the recent pause of the Janssen vaccine due to VIPIT.  This information appears under the subheading “Vaccine safety” in the revised manuscript in blue text.

- Some more data from minorities across the world and the comparison with the US data would be also helpful

Authors’ response

We agree with the reviewer and have added information to address the reviewers’ comments that appears in blue text of the revised manuscript.

-Please remove the parenthesis ‘(Figure 1)’ from the abstract.

Authors’ response

We agree with the reviewer and we have remove the parenthesis around the word “Figure 1” in the abstract.

Round 2

Reviewer 1 Report

Estimated Authors,

I've appreciated the efforts to cope with my previous recommendations and requests. In fact, most of the shortcomings I've previously noticed have been removed. Still, I've some minor requests before I could endorse the eventual acceptance of this paper, most of them relate on the style/phrasing of the introduction.

More precisely (please be aware that, as my copy lacks the row number, I would refer to the consecutive row number since the inception of the chapter + page):

  1. Page 1 - Introduction row 4 "... they [i.e. minorities] are also among the most vulnerable..." to what? I guess to "infectious diseases like SARS-CoV-2, but please revise.
  2. Page 1 - Introduction row 4 to 8 - the sections on the VH are substantially duplicated of the initial sentences of the section specifically focusing on VH. Please choice to preserve either this section or the following one.
  3. Page 1 - introduction row 9 - 12: this section is somewhat isolated inbetween two sections dealing with the problems of minotiries. It is very important but please move either at the beginning of the chapter (e.g. reporting on the severe impact of SARS-CoV-2 on US population, and then on the availability of vaccines, then shift to minorities and their issues)
  4. Table 1 - please include refereces to the studies the table refers to
  5. Page 4, Section 3.4, 6th row before the end of the section: "... after careful review by the FDA and CDC." The sentence includes a full stop that is not properly fitting with the meaning of the section. Please revise.

Author Response

May 6, 2021

Editor and Chief
Journal Vaccines

Manuscript ID: vaccines-1181419

Dear Editor,

My responses to reviewers comments regarding manuscript Manuscript ID: vaccines-1181419 entitled “Targeting COVID-19 Vaccine Hesitancy in Minority Populations in the US: Implications for Herd Immunity” are enclosed.

Thank you for giving me the opportunity to submit this manuscript to your journal for publication.

Kind regards,

Donald J. Alcendor, Ph.D.

Associate Professor

Meharry Medical College

Center for AIDS Health Disparities Research

& Department of Microbiology and Immunology

& Obstetrics and Gynecology

Hubbard Hospital 5th Floor Rm. 5025

1005 Dr. D.B. Todd Jr. Blvd.

Nashville, TN 37208

Phone: 615-327-6449

Fax: 615-327-6929

Email: dalcendor@mmc.edu

Associate Professor Adjunct

Department of Pathology, Microbiology and Immunology

Vanderbilt University Medical Center

Reviewer 1

Comments and Suggestions for Authors

Comments and Suggestions for Authors

Estimated Authors,

I've appreciated the efforts to cope with my previous recommendations and requests. In fact, most of the shortcomings I've previously noticed have been removed. Still, I've some minor requests before I could endorse the eventual acceptance of this paper, most of them relate on the style/phrasing of the introduction.

More precisely (please be aware that, as my copy lacks the row number, I would refer to the consecutive row number since the inception of the chapter + page):

Page 1 - Introduction row 4 "... they [i.e. minorities] are also among the most vulnerable..." to what? I guess to "infectious diseases like SARS-CoV-2, but please revise.

Authors’ response

We agree with the reviewer comments and have made changes to the revise manuscript in red text.

Page 1 - Introduction row 4 to 8 - the sections on the VH are substantially duplicated of the initial sentences of the section specifically focusing on VH. Please choice to preserve either this section or the following one.

Authors’ response

We disagree with the review because the contents of VH is expanded upon in the section designated for vaccine hesitancy as opposed the brief language used in the introduction. However, we have modified the language in the Introduction to address the reviewers’ concerns.

Page 1 - introduction row 9 - 12: this section is somewhat isolated inbetween two sections dealing with the problems of minotiries. It is very important but please move either at the beginning of the chapter (e.g. reporting on the severe impact of SARS-CoV-2 on US population, and then on the availability of vaccines, then shift to minorities and their issues)

Authors’ response

We agree with the reviewer and have made modification to the Introduction in red text to avoid the section being isolated.

Table 1 - please include references to the studies the table refers to

We disagree with the reviewer in that the references list for the phase III clinical trial are listed in the section for the different vaccines that includes the number of participants, efficacy data etc.

Page 4, Section 3.4, 6th row before the end of the section: "... after careful review by the FDA and CDC." The sentence includes a full stop that is not properly fitting with the meaning of the section. Please revise.

Authors’ response

We agree with the reviewers’ comments and have revised the sentence that appears in red text in the revised manuscript.